# Sensory Health and Universal Health Coverage in Canada—An Environmental Scan

**DOI:** 10.3390/healthcare12232475

**Published:** 2024-12-06

**Authors:** Hanna Asheber, Renu Minhas, Ved Hatolkar, Atul Jaiswal, Walter Wittich

**Affiliations:** 1DeafBlind Ontario Services, Newmarket, ON L3Y 3E3, Canada; asheber.hanna@gmail.com (H.A.); hatolkarved@gmail.com (V.H.); 2School of Optometry, Université de Montréal, Montreal, QC H3T 1P1, Canada; ajaiswal@perleyhealth.ca (A.J.); walter.wittich@umontreal.ca (W.W.); 3Perley Health, Ottawa, ON K1G 5Z6, Canada

**Keywords:** environmental scan, vision care, hearing care, deafblindness, dual sensory loss, sensory health, universal health coverage

## Abstract

**Background/Objectives**: The World Federation of the Deafblind Global Report 2023 reports that many countries do not have a comprehensive identification, assessment, and referral system for persons with deafblindness, a combination of hearing and vision loss, across all age groups and geographic regions. The scan seeks to inform researchers, policymakers, and community-based organizations about the status of and gaps in sensory healthcare initiatives in Canada, with the intent to raise awareness to enhance the integration and coordination of eye and ear care services. **Methods**: We conducted an environmental scan of Canada’s healthcare system and current public health policies addressing vision and hearing care in Canada at the federal and provincial levels. The scan was conducted using published literature searches from five scientific databases—Embase, Medline, PsycINFO, PubMed, and CINAHL—in combination with the gray literature review of federal, provincial, and territorial governments and non-profit organizations’ websites from April 2011–October 2022. Out of 1257 articles screened, 86 studies were included that met the inclusion/exclusion criteria. In total, 13 reports were included in the gray literature search, with 99 total articles used in the analysis. **Results**: The thematic findings indicate stigma and discrimination toward individuals with disabilities and marginalized communities (Indigenous people, rural communities, recent immigrants, people of older age, and people with disabilities), including hearing, vision, or dual sensory loss, persist. Barriers to vision and hearing healthcare access include inadequate policies, underinvestment in vision and/or hearing services, limited collaboration and coordinated services between hearing and vision services, discrepancies in insurance coverages, and lack of health system support. **Conclusions**: This scan demonstrates the persisting barriers to vision and/or hearing services present in Canada, stemming from inadequate policy and limited service coordination. Future work to address gaps, evaluate public education, and develop integrated sensory healthcare initiatives to enhance coordinated eye and ear care services, as recommended in the WHO Report on Hearing and Vision, is imperative.

## 1. Introduction

Over 1.5 billion people currently experience hearing loss, with projections indicating a rise to 2.5 billion by 2050. Similarly, vision impairment affects over 2.2 billion people globally. A unique group within these populations includes individuals with combined hearing and vision loss, often referred to as deafblindness or dual sensory loss, representing between 0.2% to 2% of the world’s population [1]. In Canada, approximately 1.1 million individuals may experience deafblindness or dual sensory loss [2]. Despite the significant prevalence of deafblindness, it is often overlooked, and many countries, including Canada, lack a comprehensive, identification, assessment, and referral system for his population across all age groups and geographic regions [3].

The WHO reports on Hearing (2021) and Vision (2019) emphasize the need for a public health approach to comprehensive vision and hearing care, given the increasing global burden of these impairments [4,5]. Universal health coverage (UHC) is crucial for providing essential healthcare services globally, offering financial protection and person-centered care [6]. Aligned with the 2030 Agenda for Sustainable Development, UHC aims to leave no one behind, as declared by all United Nations member states, including Canada. Each nation’s progress toward UHC varies based on factors such as population needs, resources, and political context. Political commitment and leadership are essential for achieving UHC and the Sustainable Development Goals (SDGs) by 2030. Canada’s UHC system faces unique challenges, including demographic aging, cultural diversity, and Indigenous rights [6].

Despite international efforts, stigma and discrimination against individuals with disabilities, including hearing, vision, or dual sensory loss, persist. Barriers to healthcare access include inadequate policies, lack of accessible materials, and limited training for healthcare professionals [7]. Disability inclusion is essential for achieving UHC, supported by Article 25 of the UN Convention on the Rights of Persons with Disabilities. However, many countries still lack comprehensive services for the deafblind population, prompting WHO initiatives to address these gaps.

The purpose of this environmental scan is to review Canada’s healthcare system and current public health policies addressing vision and hearing care in Canada at the federal and provincial levels. The findings will assess how well Canada aligns with the recommendations in the WHO reports on Hearing 2021 and Vision 2019, facilitating future planning and decision making for individuals with deafblindness. This report will also aim to raise awareness, engage stakeholders, and empower community-based organizations focused on building effective sensory healthcare plans for Canadians. It seeks to better enhance the integration and coordination of dual sensory health services across Canada by highlighting the importance of interdisciplinary and intergovernmental collaboration in administering, delivering, and reconfiguring public healthcare. The scan specifically aims to answer the following questions:
What is Canada’s current provision of hearing and vision care under the universal healthcare system at the federal, provincial, and territorial levels?What are the gaps/challenges in the care system in meeting the WHO recommendations on vision and ear health in Canada?

The findings indicate significant gaps in the provision of vision and hearing care services under Canada’s universal healthcare system. Despite existing services, key deficiencies include inconsistent access to care, insufficient data collection, and disparities in coverage and accessibility, which hinder Canada’s ability to meet the WHO recommendations. Marginalized communities (Indigenous people, rural communities, recent immigrants, people of older age, and people with disabilities) are particularly affected by these shortcomings in care.

## 2. Methods

An environmental scan is a systematic process used to gather, analyze, and synthesize information from a wide range of sources to identify trends, gaps, and key insights relevant to a specific research question or field of interest [8]. It involves reviewing both published and gray literature, such as reports, policy documents, and other non-peer-reviewed sources, to provide a comprehensive overview of the current state of knowledge, practices, or conditions within a given area [9].

For this study, we chose to use an environmental scan as the most accurate and efficient methodology to represent all available data on vision and hearing care services in Canada. By reviewing a broad spectrum of sources, including governmental reports, healthcare policies, and other gray literature, we were able to capture the full extent of services and programs across the country, providing a comprehensive understanding of current practices and existing gaps in care. This contrasts with scoping reviews as we examined unpublished literature and publicly available information to support our analysis. This diversity of information allows for effective planning and program implementation across a variety of sectors to assess the external and internal environments of health programs.

The environmental scan was conducted using published literature searches from five scientific databases—Embase, Medline, PsycINFO, PubMed, and CINAHL—combined with the gray literature review of federal, provincial, and territorial governments and non-profit organizations websites through Google search(s) from April 2011 to October 2022. Data were extracted and analyzed thematically to describe vision and hearing care services across provinces and identify gaps in service delivery in Canada. Out of 1257 articles screened, 86 studies met the inclusion/exclusion criteria. Additionally, 13 reports were included from the gray literature search. The scan involved a thorough examination of the existing eye and ear care policies at the federal, provincial, and territorial levels in Canada utilizing a variety of research methods, including literature searches and gray literature reviews. The key components of the scan included:**Review and Analysis of Publicly Available Materials**: Examination of official documents, reports, and policies related to eye and ear care at various governmental levels.**Literature Search**: Utilization of published literature searches from five scientific databases and a gray literature review of federal, provincial, and territorial governments and non-profit organization websites from April to October 2022.**Description of Provincial and Territorial Sensory Care Services**: Overview of services available in each province and territory, including utilization levels and the burden of sensory care expenses, particularly out-of-pocket costs.

An experienced research librarian from the University of Western Ontario assisted in developing the literature search strategy for each of the five databases, incorporating three key concepts: (1) vision or hearing impairment or loss, (2) universal health coverage or vision or hearing care, and (3) Canadian geography. While the same keywords were used in all five databases, each included its relevant MESH terms. Articles published from 2011 to 2022, originating from Canada, were selected. The databases searched, and the search terms used, are outlined in Appendix A.

The sources were screened in a two-stage process: title and abstract screening, followed by full-text review. The inclusion/exclusion criteria were finalized to ensure relevance to the research question and the purpose of the paper. Articles were included if they were (i) systematic or Cochrane reviews, (ii) editorials or comments on Canadian universal health coverage, (iii) public health policy reviews, (iv) studies on the utilization and effectiveness of eye and ear care initiatives, or (v) studies on other conditions causing hearing or vision loss, such as diabetes or dementia. Two team members (RM and VH) conducted screening, and disagreements were resolved by a third member (HA). The information from these sources is summarized in Appendix A.

The gray literature review aimed to highlight current benefits available to the Canadian population through government and non-governmental organization reports and address shortcomings in the Canadian healthcare system regarding universal eye and ear care. Federal, provincial, and territorial government websites were searched to identify policies and coverage for eye and ear care services in Canada. Websites of professional Canadian organizations focused on hearing and vision services, such as the Canadian Ophthalmological Society, as well as organizations catering to Canadians with vision and/or hearing loss, were also explored. Appendix A outlines and summarizes the websites searched in the gray literature review, focusing on information from April 2011 to 2022.

An examination of official documents, reports, and policies related to eye and ear care at various governmental levels was collected to prepare an overview of vision and hearing services available at the federal, provincial, and territorial levels.

The data from this environmental scan were analyzed using Braun and Clarke’s thematic analysis principles to summarize findings and produce a cohesive analysis [10]. This methodology is widely recognized for its robustness and flexibility in qualitative research, particularly when analyzing large volumes of textual data. The process involved several steps, beginning with the familiarization of the data, followed by generating initial codes and identifying potential patterns in several meetings between HA, VH, RM, and AJ. Finally, we agreed to proceed to broader themes, ensuring that these were representative of the most salient insights related to research questions. Throughout the analysis, we ensured that the themes were data-driven and reflective of the patterns that emerged from the sources, rather than being preconceived or imposed. This approach was used because of its rigorous, yet flexible nature, which allows for the identification of both explicit and nuanced patterns within the data.

## 3. Results

The data were gathered from both peer-reviewed and gray literature sources. The initial search yielded 1257 studies, with 99 meeting the inclusion criteria. Two researchers (RM and VH) conducted a full-text screening of 207 articles, with HA resolving any conflicts. The findings are summarized in a PRISMA (Preferred Reporting Items for Systematic Reviews and Meta-Analyses) flow diagram in Figure 1. The results are presented in three sections: (1) study characteristics from both peer-reviewed and gray literature; (2) provision of hearing and vision care services in Canada; and (3) gaps in these services at various levels.

### 3.1. Study Characteristics

A total of 99 articles/reports were included, the majority being peer-reviewed articles (*n* = 86) and the remainder gray literature (*n* = 13). Key information from these sources is summarized in Appendix A, respectively.

### 3.2. Vision and Hearing Care Services in Canada—Federal, Provincial, and Territorial Provision

Table 1 provides an overview of the vision and hearing services offered at federal, provincial, and territorial levels.

**Vision care**: At the federal level, Canada provides vision health benefits for those uninsured, including coverage for members of the Canadian Armed Forces, veterans, the Royal Canadian Mounted Police, as well as immigrants and refugees [16]. Most provinces and territories offer eye exams every two years for those under 18 and over 65. Regions with school- or community-based vision screening programs for children include Manitoba, British Columbia, New Brunswick, Newfoundland and Labrador, Prince Edward Island, Nova Scotia, Nunavut, the Northwest Territories, and Yukon [26,43,51,64,65,66]. However, coverage for eyeglasses generally excludes adults aged 19 and 64, who must rely on private insurance or qualify for financial assistance.

**Hearing care**: The Canadian Infant Hearing Task Force (CIHTF) rated Canada’s hearing care provision as insufficient overall. Provinces with sufficient grades included Alberta, British Columbia, Northwest Territories, Nova Scotia, Ontario, and Yukon, while the rest of the provinces received insufficient grades [67]. Coverage for hearing aids varies widely: Nunavut provides seniors over 65 with one hearing aid every five years [65], while Yukon offers coverage for children under 15 and seniors, with one hearing aid every four years [68]. Other provinces, such as Nova Scotia and Prince Edward Island, provide hearing aid subsidies for those under 21 [52,69]. New Brunswick offers partial coverage for hearing aids through the Health Services Hearing Aid Program, which assists individuals with special needs [47]. Alberta provides coverage for the purchase and/or repair of hearing aids for clients under 18, full-time students under 25, low-income individuals, or seniors [33]. Saskatchewan’s hearing aid plan offers continuous service for individuals who qualify for financial assistance [70]. In Manitoba, there is a children’s hearing aid program, as well as reimbursements of up to $2000 for seniors over the age of 65 to purchase hearing aids [28]. Quebec’s hearing devices program covers the full cost of hearing aids for all individuals under the Régie de l’assurance maladie du Québec (RAMQ) [23]. Finally, Ontario’s Assistive Devices Program covers 75% of the costs for equipment and supplies related to hearing impairments [71].

### 3.3. Gaps in Vision and Hearing Services in Canada

Gaps in vision and hearing care services are categorized into three levels: (1) the individual user, (2) the service provider, and (3) the system level.

### 3.4. Individual User Level

Thirty-two articles identified gaps at this level, including lack of awareness of available services provided, as well as stigma and barriers to access, especially among marginalized groups.

Three articles reported that barriers like a lack of knowledge of the risks of mismanaged eye care, a lack of transportation, unawareness of screening services, miscommunication by healthcare providers, stigma related to using low-vision services, and having a potential disability, deter patients from seeking care until their symptoms were unbearable [66,72,73,74].

Financial burdens also presented significant challenges, with studies highlighting reduced eye care utilization due to out-of-pocket costs (Ontario) and (Newfoundland and Labrador), particularly for those aged 20–64 [75,76,77].

Twenty articles identified that barriers were notably higher for unhoused individuals, Indigenous Canadians, low-income groups, older adults, and immigrants. For example, 20% of unhoused individuals had visual impairments, compared to just 6% of the general population [73]. Indigenous communities, particularly those in rural areas, faced multiple barriers, such as a lack of translators, social discomfort in clinical settings, high costs, and limited on-reserve care [4,78]. Immigrant Syrian children and their parents encountered cultural and linguistic challenges that impeded their access to eye care [79].

### 3.5. Service Provider Level

Gaps were identified in clinician knowledge and adherence to screening guidelines, particularly for patients with comorbidities, such as vision and/or hearing impairments. One study found that primary care providers, family physicians, and pediatricians skipped critical eye exams, leading to missed diagnoses [80]. Providers also lacked comprehensive knowledge of managing visual and hearing impairments. Collaboration between primary care providers, optometrists, and audiologists was recommended to improve screen rates [81,82]. Late diagnoses of hearing impairment were also linked to comorbidities, with 16.5% of children in a newborn hearing screening program having their hearing loss confirmed late [83]. Another study identified the need for dementia-friendly audiology due to the high prevalence of both hearing and dementia conditions within the older population [84].

### 3.6. System Level

The key gaps include underinvestment in vision and hearing services research, discrepancies in insurance coverage(s), and a lack of health system support. For instance, one study found that only 65% of Ontario children received a comprehensive eye exam by the age of 7, and only 2.3% followed the schedule for eye exams recommended by the Canadian Association of Optometrists [85]. Newfoundland and Labrador had the lowest usage of eye care services, the highest rate of visual impairment, and a shortage of ophthalmologists below the national standard, as well as a lack of provincial coverage for the cost of a routine eye exam for seniors over the age of 65 [85,86,87]. Insurance coverage disparities were highlighted, with lower-income Canadians less likely to have insurance [88]. Additionally, there is a lack of qualified ASL interpreters in hospitals for deaf Canadians [89].

### 3.7. Provincial and Territorial Scan

In terms of vision care services offered in Canadian provinces and territories, citizens under the age of 18 and over 65 are covered for eye examinations once every two years. Vision screening programs, offered in schools and through community initiatives, play a crucial role, as the Canadian Association of Optometrists reports that one in six children struggle with learning and reading due to a vision problem.

Regarding hearing care, the Canadian Infant Hearing Task Force (CIHTF) evaluated Early Hearing Detection and Intervention (EHDI) programs nationwide, assigning grades of “insufficient” or “sufficient” based on five criteria: universal newborn hearing screening, identification of babies with permanent hearing loss, intervention services including technology and communication development support, family assistance, and monitoring and evaluation of the program effectiveness [67]. However, Canada’s overall insufficient grade raises concerns about delayed diagnoses of hearing impairments in children. This delay can have severe repercussions, as children with unilateral hearing loss often struggle with academic success, behavioral issues, and speech and language development [90,91].

## 4. Discussion

Canada’s universal health coverage model, Medicare, comprises 70% public sources (General tax funds) and 30% private sources (Private health insurance and out-of-pocket payments. [92]. While Medicare provides equitable access to physicians and hospital services through thirteen provincial and territorial tax-funded public insurance plans, it is not a national system, but a collection of provincial and territorial plans, each subject to national standards [92].

In 2019, the World Health Organization (WHO) released its “Report on Vision”, calling for an integrated person-centered approach to eye care service in member states. Canada, as a signatory to the WHO’s VISION 2020 and the Global Eye Health Action Plan 2014–2019, is committed to ensuring universal access to comprehensive eye care and vision rehabilitation for all Canadians [92,93,94]. The 2019 WHO Report on Vision emphasizes integrating eye care into Universal Health Coverage (UHC), promoting interdisciplinary care, and monitoring trends in eye health services. However, Canada’s vision care services still face challenges, including a lack of research, underinvestment, and discrepancy in coverage, signaling the need for improved alignment with WHO recommendations.

Similarly, the WHO’s “World Report on Hearing 2021” outlines a public health approach to making ear and hearing care accessible for all. It calls for strengthening service delivery across all levels of care and promoting public health research, emphasizing the growing global need for hearing care and making it a public health priority [92]. While Canada offers quality audiology and speech-language pathology services, coverage for hearing aids remains limited, and the country received an insufficient grade from the Canadian Infant Hearing Task Force in 2019. Both academic literature and gray literature emphasize the need for more research, expanded coverage for marginalized groups, and a better understanding of how hearing loss intersects with other health conditions.

This environmental scan reveals a clear distinction between vision and hearing care services in Canada, with limited attention to the needs of individuals with dual sensory loss. We found no specific literature or data addressing services for the deafblind community, which presents a significant gap in understanding the healthcare needs of this population. The lack of integrated services for individuals with both vision and hearing impairments highlights a critical oversight in the healthcare system. Consequently, the implications of our findings are limited in scope, particularly in relation to the deafblind population, which remains underserved in the current healthcare system.

This scan also identified several recommendations for improving the provision of vision and hearing care services. Policymakers should reconsider the current budget allocation to fund more vision and hearing screening initiatives, which can catch impairments early and reduce long-term healthcare burdens [95]. One successful initiative is the Hearing and Otitis Program (HOP) in Inuit communities, where audiologists and trained Inuit assistants conduct school tours to detect and treat hearing impairments [96]. The program is effective because it trains local specialists and minimizes the need for patients to travel for care, which is particularly important in rural and remote regions where access to specialized care is limited [97,98].

To address gaps in hearing care, implementing a universal newborn hearing screening program is recommended to ensure the early identification of hearing loss and timely interventions [99]. Given Canada’s insufficient grade from the Canadian Infant Hearing Task Force, reassessing budgets to expand newborn hearing screening is crucial for children’s developmental success [67].

For vision care, regular eye exams should be implemented in long-term care homes to prevent vision-related impairments and improve the quality of life for older adults [100]. However, primary care physicians are advised not to routinely refer community-dwelling individuals aged 65 and older for screening, based on weak recommendations against routine screening [101,102].

A significant limitation of this scan is the limited availability of data on hearing care services and their impact on Canadians, compared to the abundance of information on vision care. Future research should explore the intersections of hearing loss with other comorbidities beyond dementia to better understand access to care in the Canadian context. Furthermore, a major limitation of this study is the absence of data specifically focused on Canadians with dual sensory loss. Our findings primarily reflect separate vision and hearing care services without addressing the unique needs of the deafblind community. This gap calls for future research to explore integrated care models for individuals with dual sensory impairments. Additionally, discrepancies in the necessity of newborn hearing screening need further exploration to develop robust recommendations. The lack of specific utilization data from the Canadian Institute for Health Information (CIHI) also limited the analysis, as this information was included in “other” category. Moreover, there is no evidence that Canada, or its provincial governments, are actively considering the recommendations from the WHO’s 2019 Vision Report and the 2021 Hearing Report. Governments are encouraged to review these recommendations and improve programming to enhance vision and hearing healthcare across the country.

Timely and appropriate vision and hearing care should be provided to all Canadians under UHC with minimal out-of-pocket costs, regardless of geographic location. Interdisciplinary referrals, expanded service coverage, early identification screenings, and increased education and awareness campaigns are essential to making progress toward UHC [71]. The WHO reports provide comprehensive guidance for expanding vision and hearing care, emphasizing regular monitoring, service utilization tracking, and assessing the burden of ear and eye diseases [103,104]. This report should be used to stimulate federal and provincial governments, advocacy organizations, and researchers to collaborate on ensuring equitable access to vision and hearing care for all Canadians—regardless of their age, socioeconomic status, or geographic location.

## 5. Conclusions

This environmental scan reveals significant gaps in the provision of vision and hearing care services under Canada’s universal healthcare system. Notably, there is a lack of integrated care for individuals with dual sensory loss, and no specific literature was found on services for the deafblind community. Despite existing services, key deficiencies persist, particularly in access, data collection, and coverage for marginalized groups. The WHO’s reports provide frameworks for improvements, and interdisciplinary approaches, expanded coverage and early identification strategies are crucial for addressing these gaps. Collaboration among stakeholders is essential to achieving truly universal vision and hearing healthcare in Canada, paving the way for an inclusive sensory health future for all Canadians.

## Figures and Tables

**Figure 1 healthcare-12-02475-f001:**
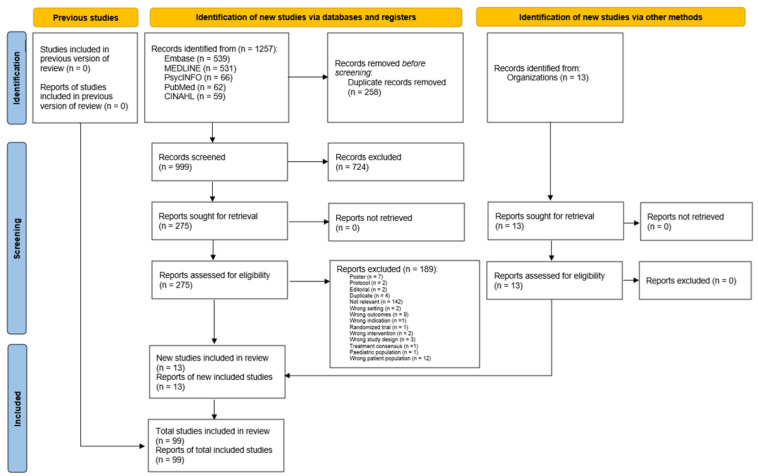
PRISMA 2020 flow diagram for updated systematic reviews which included searches of databases, registers, and other sources.

**Table 1 healthcare-12-02475-t001:** Federal and provincial hearing and vision care health policies across Canada.

Location	Vision Care Coverage	Hearing Care Coverage
Canada	**Canada Health Act**: Does not cover vision care, dental care, prescription drugs, ambulance services, and independent living [11]. **Non-Insured Health Benefits Program (NIHB)**: Covers eye exams and eyeglasses for First Nations/Inuit, Canadian Armed Forces, veterans, RCMP [12,13,14]. The Mobile Eye Clinic initiative brings optometrists to schools, youth centers, colleges/universities, seniors’ residences and community health centers to perform comprehensive eye exams [15].	Eligibility for hearing aids expanded under non-insured health benefits; coverage varies for veterans, RCMP, etc. [16].
Ontario	**Ontario Health Insurance Program (OHIP)**: covers routine eye exams once a year for ages 19 and under; 65+; conditions apply for ages 20–64 with eye disease [17,18].	**Assistive Devices Program**: covers hearing aids and equipment (FM systems) for eligible individuals [17,19]. Universal Newborn Hearing Screening [19]. **Infant Hearing Program**: universal newborn screening and assessment services [20].
Quebec	**Régie de l’assurance maladie du Québec (RAMQ)**: covers optometric services for children under 18 and seniors 65+, annually; certain low-income adults are also eligible; excludes eyeglasses, contact lenses, unless under 18 [21]. **See Better to Succeed**: offers $250 for children’s eyeglasses or contacts [22].	**Hearing Devices Program**: covers aids and assistive listening devices for all ages; no universal newborn screening [23].
Manitoba	Routine eye exam every 2 years for children under 19 and seniors 65+; coverage for medically necessary exams regardless of age [24]. **Seniors’ Eyeglass Program**: covers eyeglasses every 3 years [24,25]. **Vision Screening in Manitoba Schools**: for kindergarteners and children in Grades 1–11 offered in odd school years [26,27].	**Children’s Hearing Aid Program**: for residents under 18, covers 1 device per ear every 4 years, reimbursement of 80% for up to $500 per ear for a digital or analog programmable device [27]. **Seniors Hearing Aid Program**: provides up to $2000 for seniors 65+ [28]. Universal Newborn Hearing Screening [28].
Saskatchewan	Annual eye exam for those under 18 and diabetics [29,30]. No coverage for eyewear unless through private or other insurance [30]. Newborn screening does not include vision tests [30].	Universal newborn hearing screening [31]. Hearing tests and aids partially covered under **Family and Supplementary Health Benefits** [31].
Alberta	**Alberta Healthcare Insurance Plan (AHCIP)**: covers routine eye exams under 18 and seniors 65+; medical eye exams covered for all ages when necessary [32]. No coverage for eyewear or contact lenses for most adults [32]. Newborn screening does not include vision tests [33,34].	Hearing aids and replacement and repair of hearing aids are covered for those under 18, those between 18 and 24 in full-time post-secondary studies, and low-income seniors [33]. **Early Hearing Detection and Intervention Program** screens infants by 1 month, diagnoses by 3 months, ensures access to intervention by 6 months [32,34].
British Columbia	Routine eye exams for seniors 65+; routine eye exams once every 2 years for adults 19–64 who receive income assistance, disability assistance, or hardship assistance, or if these adults are recipients of Medical Services Only or Transitional Health Services [35]. Children under 19 in families who receive assistance obtain routine eye exams and new eyeglasses [36]. **BC Early Childhood Vision Screening** for children in kindergarten [37].	**The Medical Services Plan**: does not cover hearing aids [38]. **BC Early Hearing Program**: offers province-wide screening and early intervention, including follow-up hearing assessments and the coordination of early language services and parent support for children with a hearing loss [39].
Newfoundland and Labrador	**Income Support**: covers $55 toward the cost of a routine eye exam and a contribution toward the cost of glasses once every 12 months for children, and once every 36 months, for adults [40]. **The Eye See Eye Learn Program** offers a free eye exam and 1 free pair of eyeglasses for kindergarten children [41].	Hearing aids are not covered under the Newfoundland and Labrador Medical Care Plan [42]. Hospital Insurance covers rehabilitation services like audiology [43].
New Brunswick	**Healthy Smiles, Clear Vision Program**: for low-income children covers vision assessments and eyeglasses [44,45]. Non-covered items include eyeglasses, frames, lenses for others [46].	**The Hearing Aid Program**: for low-income individuals covers standard hearing aids and repairs [47]. Universal newborn hearing screening [48]. **Insured Hospital Services** include audiology [49].
Prince Edward Island	Social assistance applicants are eligible for exams and eyewear every 2 years. For applicants with diabetic eye disease, glaucoma, macular degeneration, or other eye conditions, more frequent and comprehensive exams may be authorized [50]. **Eye See…Eye Learn** provides one free eye exam and 1 pair of eyeglasses for kindergarten and pre-kindergarten children [51].	Hearing aids for clients up to age 21 covered; offers assessments, aid fittings, APD testing for eligible individuals [52,53].
Nova Scotia	Optometry plan covers children under 10 and seniors 65+ for routine eye exams every 2 years [54]. Comprehensive exams for children 4–19 referred by the Enhanced Vision Screening Program [54].	Hearing aids for children up to 21 years old (who are in an educational setting) are provided at wholesale cost through the Atlantic Provinces Special Education Authority (APSEA) [55]. Universal newborn screening [55]. **Nova Scotia Hearing and Speech Centers** provide services to all ages, with support for hearing aids for children up to 21 in school [56].
Yukon	**Pharmacare Program**: Covers one eye exam, new lenses, and a max contribution of $100 toward frames once every 2 years for seniors 65+ [57]. **The Children’s Drug and Optical Program**: covers 1 eye exam every 2 years, $200 for eyeglasses every 2 years and contact lenses [58]. Community health nurses conduct school-based vision testing available for children [58].	Hearing services for children and adults provided through community health nurses, including screening, hearing aid consultation, fittings, and repairs [57]. **Pharmacare Program**: Seniors 65+ qualify for 1 hearing aid in a 4-year period and repair and adjustment of 1 hearing aid with a 12-month warranty [57]. Universal newborn hearing screening [58].
Northwest Territories	**Extended Health Benefits for Seniors Program**: Provides $300 for a standard prescription or $40 for a high-index prescription every 2 years. Adults over 18 are eligible for vision care benefits every 2 years, and those under 18 are eligible every year. Covered services may include frames, lenses and contact lenses [59]. Annual vision screening for kindergarten children provided by public health units in Yellowknife [60].	**Extended Health Benefits**: partially cover hearing aids [59]. Infant hearing screening [59].
Nunavut	**Nunavut Health Care Plan**: does not cover optometric services but it covers eye exams, treatment and operations provided by an ophthalmologist [61]. **NIHB** covers exams and one pair of glasses every 2 years for adults over 19 and annually for children [62]. Vision screening at kindergarten entry [63].	**Nunavut Health Plan**: covers eligible seniors for the full cost of 1 hearing aid every 5 years [63]. Kindergarten health assessments include hearing screening, and developmental screening [63].

## Data Availability

The original contributions presented in this study are included in the article/Appendix A, further inquiries can be directed to the corresponding author.

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
