# Peer review of "Sensory Health and Universal Health Coverage in Canada—An Environmental Scan"

_healthcare, 2024, doi:10.3390/healthcare12232475_

Round 1
Reviewer 1 Report
Comments and Suggestions for Authors
This manuscript uses an environmental scan as a method to understand the current services and challenges in the Canadian healthcare system.
The methodology is appropriate and is well-described. I have one methodological question and one analytic question. First, I note that the articles/websites were collected during April to October 2022, but did the authors have exclusion criteria based on the age of the information? Information online can become out of date, so I would advise the authors to capture some information that indicates the recency of each included resource. This seems important because the goal is to understand the “current provision of hearing and vision care” in Canada. This should be noted.
The thematic analysis is appropriate and I found that the summarized information categorizing gaps at the level of the individual user, service provider, and system to be useful. But one thing that I found missing from the analysis was the accessibility of these services for people at the intersection of deaf and blind. The manuscript opens by stating that there is often no referral system for people who are deafblind. But the rest of the article examines services for those who have difficulty with hearing separately from difficulty with vision. I would recommend that the authors either incorporate this perspective into the analysis/discussion, or reframe the background to not discuss this.
Editorial note: The hyperlinks in Table 2 do not work. Also Table 2 is mislabeled. The flow diagram is Figure 1. This table needs a new title.
Reviewer 2 Report
Comments and Suggestions for Authors
Peer Review Report
Sensory Health and Universal Health Coverage in Canada healthcare-3304808
Thank you for the invitation to review the manuscript Sensory Health and Universal Health Coverage in Canada healthcare-3304808. This is a comprehensive, well-written article and I congratulate the authorship for this piece of work.
Summary
The authors have investigated Canada’s current provision of hearing and vision care at federal provincial and territorial levels and have identified gaps/challenges in the care system according to WHO recommendations on ear and vision health in Canada.
The authors have utilised an ‘environmental scan’ methodology which has involved three key components – a review and analysis of publicly available materials, a literature review of peer-reviewed scientific articles and a description of available sensory (vision and hearing) services at the provincial and territorial levels in Canada.
I commend the authors on utilising a comprehensive, systematic approach to answer the research questions. The introduction clearly explores the context of the work, which is well justified considering the gap between the global health mandate for hearing and vision policy and potential gap at the country and local levels. The rationale behind the study is well justified and laid out with the authorship expressing clearly the purpose and need of the work.
The method section is particularly well written and comprehensive in nature. The authorship have justified the three components of the environmental scan and related them back to the purpose of the study within the discussion.
The results section is very clearly laid out utilising relevant sub-headings.
The discussion does a good job of summarising the key findings within the broader context of the work.
Overall presentation is excellent, clearly and articulately written.
Minor amendments:
Line 89: Please explain in the article the “environmental scan” method. Please include the theoretical basis to this method. Provide a justification of this approach in comparison to say a scoping review methodology.
Line 142: You have utilised Braun and Clarke’s thematic analysis principles to ‘summarise findings and produce a cohesive analysis’ yet there is very little detail on how this was done. Please provide greater detail around what process was taken, how validity was ensured and how consensus was reached.
Consider Table 1 be included within an appendix
Line 140: The table 2 caption is incorrect as it states it is a PRISMA 2020 flow diagram
Figure 1: please check the number of n=99. In the abstract it looks like it was 82+13 (n=95).
Line 281- Please clarify what you mean by marginalized groups
The discussion section is well written, however there is a lack of inclusion about findings of the work in relation to deaf-blind communities specifically, an important part of the study. If no literature was found specifically related to deaf-blind communities, then this must be discussed and linked to the implications of the work.
Once again, thank you for the opportunity to review this work.
Round 2
Reviewer 1 Report
Comments and Suggestions for Authors
The authors did a thorough job of addressing reviewer comments. The updates enhance the overall presentation, particularly the context of the intersection of hearing/vision difficulty and the absence of resource in this area. I have no further comments.